# After the Crimea crisis: Employee discrimination in Russia and Ukraine

Iuliia Naidenova[1], Cornel Nesseler[2]*, Petr Parshakov[1], Aleksei Chusovliankin[1]

1 NRU HSE, Perm, Russia, 2 Norwegian University of Science and Technology, Trondheim, Norway

* cornel.m.nesseler@ntnu.no

## Abstract

This paper examines the issue of employee discrimination after a political crisis: the annexation of Crimea. The annexation, which resulted in a political crisis in Russian-Ukrainian relations, is a setting which allows us to test if a bilateral political issue caused employee discrimination. We use a quasi-experimental approach to examine how the political crisis influenced participation in major sports leagues in Russia and Ukraine. The results show that the employment conditions significantly worsened since the Crimea crisis started.

**Data Availability Statement:** All data are available from the Harvard Dataverse at DOI: https://doi.org/10.7910/DVN/LUH6HF.

**Funding:** The authors received no specific funding for this work.

## Introduction

In March 2014 the Russian government annexed the Crimean peninsula, a move widely criticized by several western governments (e.g., US, France, UK, and Germany) and the country that lost part of its territory; Ukraine. As a result of the annexation, the European Union (EU), among others, imposed economic sanctions and individual restrictive measures on Russia, Russian people, and Russian entities ([1]). The US and Canada also imposed sanctions on Russia. In contrast to the EU measures those measures are not time restricted. In addition to those measures, the Crimean peninsula was put under restricted economic relations. All sanctions were supposed to lapse in July 2015 but have been continuously extended since. The annexation had significant effects on the bilateral relations between Russia and Ukraine. The Crimean crisis resulted in a still on-going (at the time of publication) civil war in the Ukrainian Donbass region. The Russian government advertised the annexation with the slogan "Crimea is in my heart." This official government perspective is routinely supported by most of the government-controlled media and social media outlets (for the support regarding "traditonal" media outlets see [2]; for support regarding social media see [3]). From the Russian government's perspective the annexation was a natural incident or natural process that was bound to happen. The perspective of the Ukrainian government and media is profoundly different and evaluates the annexation as an attack on its sovereignty—a view supported by most western governments.

The aim of this paper is to analyze whether the annexation of the Crimea had a significant impact on the employment situation for Ukrainian and Russian employees living in the country of the other. The countries are neighbors with a long history of close economic ties, although their diplomatic relations are historically problematic ([4, 5], pp.58-95). We

**Competing interests:** The authors have declared that no competing interests exist.

empirically examine whether the Crimean crisis had a lasting impact on the participation of Ukrainian employees in Russia (and vice versa). We use professional soccer and hockey players as an example for employees living in Russia or Ukraine. Specifically, we analyze if the playing time (i.e., participation) of Ukrainian players in Russia (and vice versa) significantly changed. An analysis if players were treated or acted differently (i.e., received more penalties/ cards) was explored but we refrained from focusing on it because of unobserved confounds.

Sport provides us with a number of advantages. First, we can clearly measure the participation and the quality of each employee. We can measure the participation of the players using their playing time (soccer) or using the number of matches they played (hockey). Thus, we can clearly detect how much a club relied on an employee before and after the Crimean crisis. In many other industries individual participation and performance measurements are not observable.

Second, we can assume that the decision to hire a player or extend a contract is made by local employers. Although many sports clubs sell a product to nationwide, sometimes international, audiences the sports club has one decision-making branch clearly located in the city where the club plays. In many other industries the local branch of a company does not have the sole responsibility to hire or dismiss international employees. This is the case for many expatriates but not necessarily for sports players.

Third, in contrast to other industries, professional soccer players are highly flexible [6]. Many players change their club every two or three years to maximize their potential income. Thus, clubs and players have the possibility to change to another club or league in a matter of days, which means that both player and clubs can actively react to political changes.

The results show that the employment situation for Russian employees in Ukraine worsened after the crisis. In both observed industries (soccer and hockey) the participation of Russian employees working in the Ukraine significantly decreased. The results also show, however, that for Ukrainian employees in Russia the situation is different. The participation of Ukrainian employees decreased one industry decreased (soccer).

The paper is structured as follows: In section 2 we give an overview of the related literature. In section 3 we provide a brief historical background for the Crimean peninsula. In section 4 the empirical strategy is presented, first for soccer then for hockey. In section 5 we present and analyze the data, first for soccer then for hockey. In section 6 we discuss the results and summarize our findings.

## Literature review

In his seminal work, Becker [7] describes how discrimination has negative consequences for both employee and employer. Theoretically, to maximize profits companies hire the best possible employees with respect to wage restrictions. This process is similar in a sports environment. Clubs maximize either profit or winning percentage (cf., [8, 9]). Aggregate talent is similar to the quality of employees. This means that when clubs consciously choose not to hire players with a specific nationality they decrease their team's aggregate talent and, respectively, the leagues' aggregate talent (cf., [10]). Thus, discrimination has a negative effect on the player, club, and league. Researchers frequently examine discrimination in the sports sector.

According to Becker's theory of taste-based discrimination, "some economic actors prefer not to interact with a particular class of people and are willing to pay a financial price to avoid such interactions" ([11], p. 431). This is the case for employers when hiring a candidate but also when an employee is already working for a company. Alternative theories of discrimination, the information-based theory [11] or statistical-discrimination theory [12, 13], suppose that a firm or a group of individuals have mistaken beliefs about another group's skill level and

act accordingly. In most research on discrimination, related to gender and race discrimination, it is difficult to identify whether the discrimination is taste-based or information-based [11].

There is evidence of discrimination against foreign workers in different countries [14, 15] including Russia [16] and Ukraine [17]. The negative perception of foreigners might be amplified during or after a conflict. In both Ukraine and Russia, migration and the on-going integration of diverse foreign groups are controversial topics of discussion [18] and [19], respectively). Language is believed to be an important factor in regarding foreign workers success in integration. Ther [20] examines post-war migration in Poland and Germany. He finds that sharing a similar language is helpful but the influence of "outside powers" (e.g., the government of the migrants home country) is decisive.

In sports, the issue of discrimination is widely discussed. Even in relatively peaceful regions and times, sports clubs do not per se treat different groups equally [21]. Previous research investigates discrimination in sports by the analysis of attendance [22, 23] and broadcasting [24], which depend on the players' characteristics. For most Russian soccer clubs, revenues from ticket sales and broadcasting are relatively low in comparison with their budgets [25], nonetheless, supporters attitudes may affect the decision to sign a player. Research by Coates et al. [26] finds no evidence that Russian fans discriminate against foreign players; however, Arnold and Veth [27] describe the under-institutionalized Russian fan culture and violent behavior of supporters.

Whereas most discrimination research focuses on race, gender, or nationality, it can also be caused by negative perceptions due to specific events. Several authors show that violent attacks can have a lasting influence on the perception of different groups. Ahluwalia and Pellettiere [28] examine how the perception of Sikhs (among other minority religious groups) in the US changed after the terrorist attacks on September 11, 2001. They find a strong negative correlation. Gautier et al. [29] examine how the murder by a Muslim, of the Dutch film maker, Theo van Gogh, negatively influenced the perception of Muslims in the Netherlands. Brüß [30] examines experiences of discrimination in Madrid after the Al-Quaida underground bombing in 2004. Similar to the results from other researchers, Brüß finds that the relation between minorities (i.e., Muslims in the case of Madrid) and natives can be volatile and depends on the extent of perceived threats.

In the case of the Russia-Ukraine conflict, the decrease in demand of each other's players cannot be due to the information-based theory but possibly related to tastes of soccer clubs' management, fans, or players. Especially, consumers (i.e., fans) might play a prominent role as discrimination cannot only be related to the employers' tastes, but also to the customers' tastes. During the Crimean crisis the relationship between the Ukraine and Russia became increasingly violent. Therefore, it could be reasonable for Russian clubs to discriminate some players to avoid clashes between fans and prevent offensive behavior against players. The Russia-Ukraine case is especially interesting because the languages are similar but government relations are far from usual.

## Concise historical background

Crimea is a peninsula in the Black sea, located at the southeastern part of Ukraine connected to the mainland region Kherson. East of Crimea is the Russian federal subject Krasnodar Krai. The peninsula is around 27,000 km$^2$.

In 1783 the Russian Empire annexed the Crimea from the Crimean Khanate. The Crimean Khanate was a vassal state from the Ottoman Empire, which existed until 1922. After annexation by the Russian Empire "the Russification of the Crimean population has been carried out through massive resettlement of ethnic Russians of already Russified subjects from central and

northern Russia" ([31], p. 37). This led to a complete change of the demographic constellation because Tatar, Greek, and Bulgarian minorities were deported. In 1922 Crimea was incorporated into the newly founded USSR as an autonomous state. After WWII Crimea lost its autonomous status and became an average administrative region (viz., oblast) ([31], p. 39). Between 1941-1944 Crimea was occupied by the German Reich. In 1954 the Crimean region was transferred within the Federal Republic from Russia to Ukraine. Saluschev ([32], p.38) writes, "it was an insignificant event as even a thought of the Soviet Union's eventual implosion was unthinkable."

After the dissolution of the Soviet Union in 1991, Crimea remained an autonomous state within the jurisdiction of Ukraine. This status was supported by a referendum in December 1991 [33]. In 1992 the Crimean government, lead by the pro-Soviet Republican Movement of Crimea, declared independence, which was later annulled by the Ukrainian government. As a result the government "agreed to strengthen Crimea's autonomous status." ([31], p.39). After the dissolution of the Soviet Union Crimea faced a unique position compared to other Ukrainian regions. "Crimea is the only Ukrainian region with an ethnic Russian majority." ([34], p.3) Crimea also inherited a Soviet socio-economic structure which invokes symbolic, literary, and historical memories.

Russian separatist movements were active in Crimea since the unsuccessful independence declaration in 1992 [31, 34]. The Ukrainian government knew about this ongoing process and "warned the West in 2007 of Russia's policy of destabilizing the Ukrainian government, particularly in Crimea" ([31], p.39-40). In 2014 protests that complained about the Ukrainian political and economic agenda became frequent. The protests and the reaction of the government became increasingly violent. Although the EU-brokered an agreement between the protesters and the government, the Ukrainian president Yanukovich fled Ukraine in February, 2014, for unknown reasons [35]. The Russian government saw an opportune moment to seize Crimea (with the help of local protesters and forces). Bebier ([31], p.41) writes, "The military take-over of Crimea was obviously well-prepared, rehearsed in advance and professionally executed." The seizure was legitimized by a referendum on the Crimean peninsula in March 2014. 96.77% voted in favor of seceding from Ukraine. Regarding the referendum's legality, Marxen ([36], p.382) states "that holding the referendum as such did not violate international law, but that it did not comply with international standards in regard to its modalities." The United Nations declared that the referendum "has no validity and cannot form the basis for any alteration of the status of Crimea." [37] Thus, whether Russia is an occupying power in the Crimea is disputed. Fig 1 gives a concise overview about the crisis between November 2013 until April 2014.

Between 8-10 people were killed during the annexation of the Crimea. The uprising in Crimea, however, had a close connection and followed a similar course (pro-Russian demonstrations which led to violent conflicts) to the war in the Donbass region in Donetsk and Luhansk between Russian forces (and Russian forces inside Ukraine) and Ukrainian forces [38]. Furthermore, the Crimean crisis could be evaluated as the trigger for the Donbass war. The war in the Donbass region led to more than 10,000 casualties [39]. Additionally, real GDP dropped more than 19% from 2014 to 2016 [40].

## Empirical strategy and estimations

Our identification strategy is based on a difference-in-differences approach. Difference-in-differences is a technique which is widely used in social sciences and tries to mimic an experimental research design using observational study data. It examines the effect of a treatment on an outcome by comparing the average change over time in the outcome variable for the

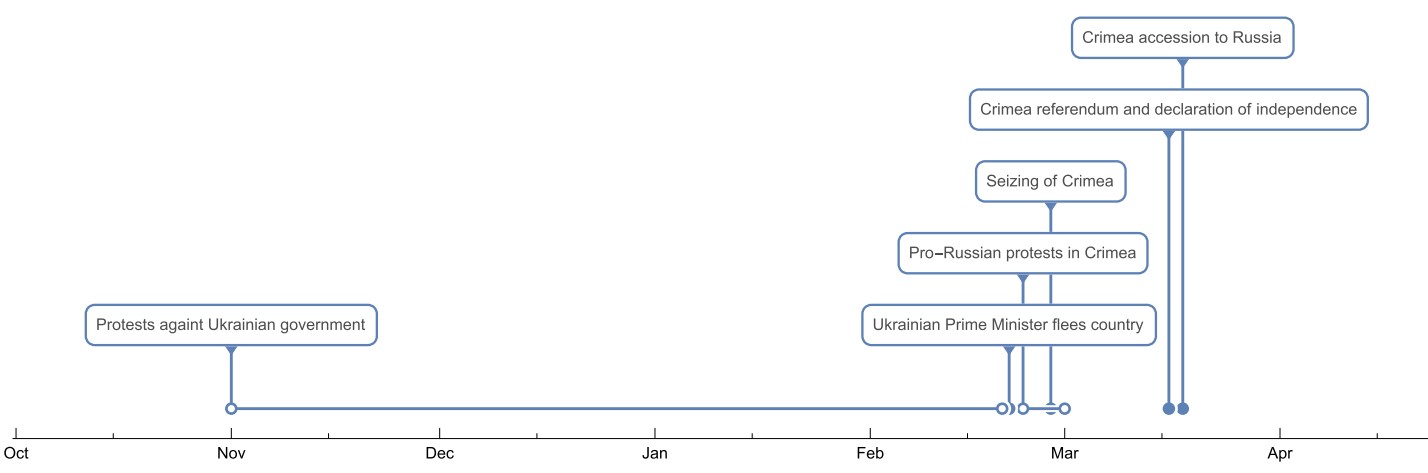

**Fig 1. Timeline of the Crimean crisis between November 2013—April 2014.**

treatment group to the average change over time for the control group. In our cases, the treatment is the political crisis caused by the annexation of the Crimea. The treatment group, affected by the political crisis, includes Russian employees in Ukraine and vice versa. This treatment is a completely exogenous shock for both Russian and Ukrainian soccer clubs. The difference-in-differences approach assumes that without the treatment, the treated group would show a trend similar to that observed for the control group [41]. The choice of the control group is crucial. To check the robustness of the results, we conduct the analysis using two options. First, we use as a control group all players in the corresponding league except the treated ones. Second, we use as a control group only foreign players in the corresponding league except the treated ones. To test the effect of the Crimean crisis on Ukrainians in the Russian league, the control group consists of all foreign players in the league except Ukrainians (and vice versa). The second control group formation assumes similar trends in participation and behaviour for foreign players in the league, whereas for domestic players the trend can be different. When we use a subsample without domestic players, we have a lower number of observations. Thus, the percentage of treatment group (Russian employees in Ukraine and vice versa) is higher.

## Soccer

In the following section we describe the econometric strategy of our approach. Soccer is the most popular sport in both Ukraine and Russia. Both leagues compete with their best teams in international European competitions (i.e., Champions League and Europa League).

## Econometric strategy

The approach for Ukraine and Russia is the same. This means that Models 1 and 3 are for the Russian league and Models 2 and 4 for the Ukrainian league:

$$log(minutes_{it}) = \quad \alpha_0 + \alpha_1 \cdot UKR_i + \alpha_2 \cdot crisis_t + \alpha_3 \cdot Distance_i + \alpha_4 \cdot UKR_i \cdot crisis_t +$$

$$\alpha_5 \cdot UKR_i \cdot crisis_t \cdot Distance_i + \mathbf{X} \cdot \beta + \mathbf{W} \cdot \delta + \sum_{s=1}^{8} \gamma_s \cdot season_s + \epsilon_{it} \quad (1)$$

$$log(minutes_{it}) = \alpha_0 + \alpha_1 \cdot RUS_i + \alpha_2 \cdot crisis_t + \alpha_3 \cdot RUS_i \cdot crisis_t + \mathbf{X} \cdot \beta +$$

$$\mathbf{W} \cdot \delta + \sum_{s=1}^{8} \gamma_s \cdot season_s + \epsilon_{it} \tag{2}$$

In Eqs 1 and 2, the dependent variable $minutes_{it}$ is the log of the average playing time of player $i$ during a season $t$. Playing time is used to analyze how much a club needs a certain player. It is an important indicator for club and player. When a club values a player it will try to maximize the player's effort and playing time. Additionally, for a player, a longer playing time might be an indicator for a longer professional career (cf., [42]). Several papers see playing time as the performance characteristic and analyze if physiological characteristics (e.g., velocity) constantly vary during playing time (see e.g., [43, 44]). Becker ([7], pp. 39-50) describes various ways how employers can discriminate against employees (e.g., depending on the employers taste or the amount of competition in the industry). In this case the employer can choose to give some employees a lower playing time and, thus, satisfy e.g., the employers taste. Research does not provide, however, a clear statement regarding the effect of diversity on performance, compare e.g., Maderer, 2014 [45] and Prinz and Wicker, 2016 [46].

The Russian government stressed on many occasions that the annexation was a natural process or incident. They explain this process either because of the large share of the Russian population and the historical connection to Russia. Thus, after the annexation they could expect Ukrainians to behave differently but Russians. This natural process, as described by the Russian government, was not supposed to be followed by any behavioral change towards Ukrainians. It is important to stress that any behavioral change within Russia would run counter to this argument.

We use dummy indicators for being Ukrainian $UKR_i$ or being Russian $RUS_i$ in the Russian and the Ukrainian league, respectively. $crisis_t$ is a binary indicator of post-crisis period. $UKR_i \cdot crisis_t$ is an interaction term for being Ukrainian in the post-crisis period. Finally, we include the distance for every Russian club to Kerch (a Crimean city). Using another Crimean city, e.g., Sevastopol does not yield statistically significant different results. With an increasing distance the involvement in the Crimea crisis might decrease, therefore we include an interaction effect for the distance to Crimea and being Ukrainian after the beginning of the political crisis. Distance might be a pivotal factor as the distance to Crimea greatly differs between Russian clubs.

The model for the Ukrainian league is the same as Model 1 with a dummy indicator for being Russian instead of being Ukrainian and the corresponding interaction effects. However, we do not include the distance to Crimea for Models 2 and 4.

The vector $\mathbf{X}$ includes information about employees (players). Employees in companies differ, therefore we want to distinguish between various characteristics. We include an estimate of a players current transfer value from the transfermarkt.com. This is a crowd-sourced metric for player market value. The advantage of this variable compared to fixed effects is that it varies from season to season. Such metrics are often used to control for individual or team skill [47–50]. Market value includes, among others, performance measurements and the age of a player. In addition, it reflects the future prospects of a player, the real demand in the market, or the actual paid transfer fees. Other factors, such as prestige and marketing possibilities, are also incorporated. Nonetheless, the market value is an estimate and does not accurately show the value of a player. We use the market value because it gives an overview about the strength of a player and sets the strength in relation to other players from the team and the league.

Some players are too young or are previously unknown to the public and have a market value of 0. We control for those players because they could distort the results. We include the

age of the player to control for life cycle effects. We include the age of the player and the age[2]. We include binary variables for the position of a player: keeper, forward, midfielder, or defender. The position might have an effect on the general playing time and thus bias the results (e.g., keepers receive fewer cards than players on other positions). To control for the fact that the demand on playing time of a player is bounded by the total time of all games, we include the dummy for players who participated 90 minutes in all games in a season. Finally, we capture whether a player moved to another team within the season. Moving to another team could significantly influence the playing time.

The vector **W** contains information about club features, namely, a club's position in the table in the current season, distance to Crimea, and club's fixed effect. $season_s$ are season dummies to control for unobservable season effects. $\epsilon_{it}$ is a random error term. We cluster standard errors over the clubs, and control for the years in the regression. Time matters in terms of standard errors, but since we can't cluster both over season and club we cluster over the club, because it is necessary to account for club-specific variation. We do not use a random effects estimator, as its assumptions do not hold. Instead we include controls for player quality, club, season, and position effects.

It should be noted, that the political conflict might affect the number of Ukrainians in Russia (or vice versa), which could affect our results. Ideally, there should be no roster change before and after the treatment. Unfortunately, this is not our case. We believe that since we use individual-level data, we observe those players who decided to stay or were forced to stay by their contract. The decision to stay on the team was already made, and in our regressions we estimate if their playing time is reduced.

Table 1 provides an overview of the data used in this paper. Our dataset includes 696 players (2,935 observations) from the Russian Soccer Premier League and 550 players (2,770 observations) from the Ukrainian Premier League. One observation is one player ($i$) in one year ($t$). Our sample includes all players—not only those who appeared both before and after 2014.

## Hockey

Hockey is the second most popular sport in the Ukraine and Russia. Especially in Russia, the league attracts national players from around the world. Our dataset includes 1,154 players (4,730 observations) from the Russian hockey clubs, and 442 players (725 observations) from the Ukrainian Hockey League. During and after the Crimean crisis the Ukrainian Hockey League was severely affected by the war in the Donbass region. Several teams stopped playing and only returned in 2015. Table 2 shows summary statistics for hockey players.

## Econometric strategy

The econometric strategy for hockey is similar to the strategy for soccer. The difference is in the choice of player participation and penalty metrics. The most important difference is that hockey statistics for players in the Ukraine and Russia are reported only if they play in a match; however, how many minutes participants play in a match are not reported. Thus, in contrast to the soccer analysis we must limit the hockey analysis to the number of games played. The number of clubs in Ukraine before and after crisis has changed. It was nine clubs before and eight after. The number of clubs is almost the same, but some of them changed their location. The share of Russian players is different for the clubs which remain in the league compared to those clubs that left, but this difference is not statistically significant.

In contrast to the data from soccer, we do not have the players market value in the vector **X**. To accurately assess the performance or the skill level of the players we control for the goals they scored, how many assists they gave, and the plus-minus. The plus-minus measures the

**Table 1. Summary statistics for Russian and Ukrainian soccer leagues.**

| Variable | Mean | Std. Dev. | Min. | Max. |
|---|---|---|---|---|
| *Russian premier league (n = 2,935)* | | | | |
| Average minutes per game * | 32.64 | 26.08 | 0.02 | 90 |
| Market value in million euro | 2.16 | 3.52 | 0 | 43.2 |
| Zero market value | 0.05 | 0.22 | 0 | 1 |
| Age | 25.47 | 4.19 | 15 | 42 |
| Ukrainian player | 0.02 | 0.12 | 0 | 1 |
| Russian player | 0.62 | 0.49 | 0 | 1 |
| Other nationality | 0.36 | 0.44 | 0 | 1 |
| Player is defender | 0.31 | 0.46 | 0 | 1 |
| Player is keeper | 0.12 | 0.32 | 0 | 1 |
| Player is midfielder | 0.42 | 0.49 | 0 | 1 |
| Player is forward | 0.15 | 0.36 | 0 | 1 |
| Distance to Crimea (th. km) | 1.20 | 0.69 | 0.2 | 3.54 |
| Team change during season | 0.27 | 0.44 | 0 | 1 |
| Season | 2013.25 | 2.05 | 2010 | 2016 |
| Team** | 17.87 | 10.65 | 1 | 35 |
| *Ukrainian premier league (n = 2,770)* *** | | | | |
| Average minutes per game * | 34.11 | 24.57 | 0.03 | 90 |
| Market value in million euro | 1.09 | 2.08 | 0 | 21.6 |
| Zero market value | 0.14 | 0.34 | 0 | 1 |
| Age | 24.62 | 4.24 | 15 | 41 |
| Ukrainian player | 0.73 | 0.45 | 0 | 1 |
| Russian player | 0.02 | 0.12 | 0 | 1 |
| Other nationality | 0.25 | 0.36 | 0 | 1 |
| Player is defender | 0.31 | 0.46 | 0 | 1 |
| Player is keeper | 0.11 | 0.31 | 0 | 1 |
| Player is midfielder | 0.42 | 0.49 | 0 | 1 |
| Player is forward | 0.15 | 0.36 | 0 | 1 |
| Team change during season | 0.24 | 0.42 | 0 | 1 |
| Season | 2013.24 | 2.00 | 2010 | 2016 |
| Team** | 19.09 | 10.20 | 1 | 36 |

NOTE.

* The number of teams which compete every year in the Russian and the Ukrainian league is different and, accordingly, the possible maximum playing time. We use playing time per game corrected for injuries.

** Due to relegation to a lower league the teams who play in the highest league change every season.

*** In 2014 the league format in Ukraine changed from 16 to 14 teams and again in 2016 from 14 to 12 teams.

difference between the goals a team scores and the goals a team receives during the time a player is on the pitch. Again, we include the age and the position of the player. Players in hockey are categorized into four different positions (goalkeepers, defenders, forwards, and undeceive). In line with the previous soccer analysis, club dummies (the vector **W**) and season dummies are included.

## Empirical results

Fig 2 is an overview of the results without the inclusion of covariates. The y-axis shows the minutes played per season in the leagues in Ukraine and Russia. The two graphs at the top of

**Table 2. Summary statistics for Russian and Ukrainian hockey leagues.**

| Variable | Mean | Std. Dev. | Min. | Max. |
|---|---|---|---|---|
| *Russian Superleague (n = 4,730)* | | | | |
| Matches played * | 31.61 | 18.46 | 0 | 61 |
| Age | 26.29 | 4.86 | 17 | 42 |
| Ukrainian player | 0.01 | 0.13 | 0 | 1 |
| Russian player | 0.69 | 0.46 | 0 | 1 |
| Other nationality | 0.30 | 0.46 | 0 | 1 |
| Goals scored | 4.28 | 5.35 | 0 | 48 |
| Assists | 6.38 | 7.28 | 0 | 60 |
| Plus Minus | 0.47 | 7.5 | -27 | 46 |
| Player position | 2.97 | 1.14 | 1 | 4 |
| Distance to Crimea (th. km) | 2.20 | 1.63 | 0.32 | 7.26 |
| Season | 2013.19 | 2.03 | 2010 | 2016 |
| Team ** | 12.80 | 7.88 | 1 | 25 |
| *Ukrainian Hockey League (n = 560)* | | | | |
| Matches played * | 18.18 | 12.12 | 0 | 42 |
| Age | 24.1 | 6.21 | 15 | 49 |
| Ukrainian player | 0.83 | 0.37 | 0 | 1 |
| Russian player | 0.1 | 0.3 | 0 | 1 |
| Other nationality | 0.17 | 0.37 | 0 | 1 |
| Goals scored | 4.05 | 6.18 | 0 | 45 |
| Assists | 6.24 | 8.84 | 0 | 61 |
| Plus Minus | -1.3 | 5.64 | -43 | 27 |
| Player position | 3.25 | 0.97 | 1 | 4 |
| Season | 2013.6 | 2.07 | 2010 | 2016 |
| Team ** | 8.63 | 4.20 | 1 | 15 |

NOTE.

* The number of teams which compete every year in the Russian and the Ukrainian league is different and, accordingly, the possible maximum matches played.

** Due to relegation to a lower league the teams who play in the highest league change every season.

the figure show the results for the soccer leagues, the two graphs at the bottom show the results for the hockey leagues. Fig 2 shows that the number of Russians in the Ukraine soccer league decreased since 2015. In all other leagues and cases, however, neither Ukrainians nor Russians suffer from apparent discrimination. Thus, in the following subsection we analyze how the inclusion of covariates influences the results.

## Soccer

Table 3 shows the regression estimates. We compare the playing time of Ukrainians to foreigners in Russia (Model 1) and Russians to foreigners in Ukraine (Model 2). In Model 3 we combine Russians and foreign players into one control group. In Model 4 we combine Ukrainians and foreign players into one control group. Models 1 and 3 are for the Russian league, Models 2 and 4 for the Ukrainian league. The intuition for the identification of a control group in Models 1 and 2 is that different reasons might affect the minutes played by Russian players and foreign players. For example, the Russian Premier League has a quota system for the foreign players: the minimum number of Russian players on the pitch is seven (it was six until

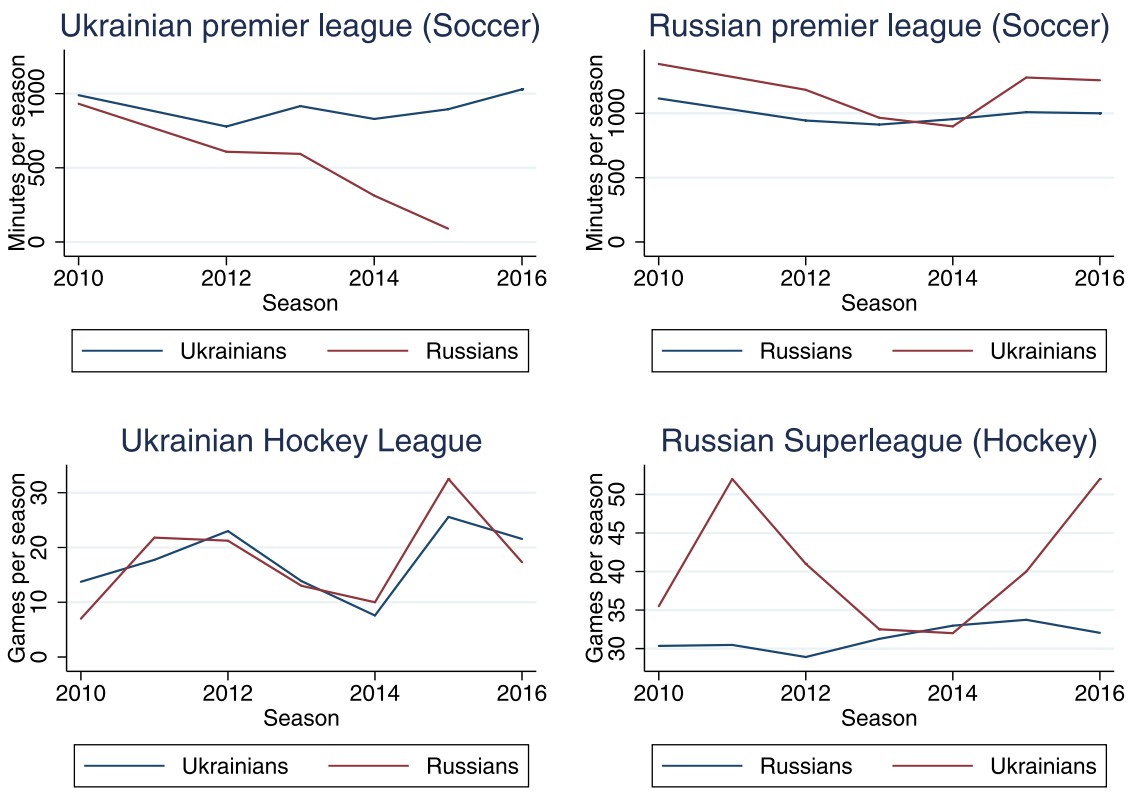

**Fig 2. Participation overview.**

2012). So, the joint sample of Russian and foreign players might be an unreliable control group to test the discrimination of Ukrainian players. However, Models 3 and 4 are used as a robustness check. Thus, to capture political discrimination we should compare e.g., Ukrainian players with Kazakhstan players, but not with Russian players.

Model 1 (Table 3) shows that Ukrainian players played statistically significantly less after the crisis. Moreover, with increasing distance to Crimea, Ukrainian players are less likely to play less. Since our dependent variable is in logs, we evaluate marginal effects as $(e^{\beta}-1) \cdot 100$ to determine the percentage change in the dependent variable. Taking into account the coefficients for the first two variables and the average distance to Crimea, the marginal effect is -52% (comparing this to other foreign players, which indicates how tied the leagues were before the crisis). Moreover, for the longest distance (7.26 th. kilometers) the effect of the political crisis is almost zero. The size of the effect is robust to exclusion of distance to Crimea. We have estimated regression without this variable and its interaction and the effect is -46%. We have added this regression to the S1 Table. The distance to Crimea is statistically significant. The effect is negative. This means that the average player of a northern team plays less. As we capture time-constant club characteristics with club-level dummies, it seems that some time-variant characteristics of clubs are correlated with the distance from Crimea. The distance from Crimea is not correlated with the clubs' proxy of the budget; total market value.

The size of the effect is higher than in other studies focusing on discrimination. The most relevant studies, however, consider racial discrimination or discrimination against migrants. In our case, relations between Russia and Ukraine deteriorated sharply and rapidly. The social consequences of the crisis were also widely discussed in the media. For instance, articles

**Table 3. Russian and Ukrainian soccer.**

| | Dependent variable: log(average minutes) | | | |
|---|---|---|---|---|
| **Model** | **(1)** | **(2)** | **(3)** | **(4)** |
| **League country** | **Russia** | **Ukraine** | **Russia** | **Ukraine** |
| **Control group** | **Foreigners** | **Foreigners** | **All players** | **All players** |
| Ukrainian player | 0.152 | | 0.153 | |
| | (0.182) | | (0.202) | |
| Ukrainian player · After crisis | -1.074** | | -0.880* | |
| | (0.502) | | (0.459) | |
| Ukrainian player · After crisis · Distance to Crimea | 0.279* | | 0.210* | |
| | (0.161) | | (0.122) | |
| Russian player | | -0.227 | | -0.214 |
| | | (0.147) | | (0.151) |
| Russian player · After crisis | | -0.572* | | -0.527 |
| | | (0.299) | | (0.315) |
| After crisis | -0.015 | 0.107 | 0.012 | 0.261 |
| | (0.149) | (0.177) | (0.113) | (0.123) |
| Distance to Crimea | -0.210*** | | -0.107* | |
| | (0.057) | | (0.048) | |
| Age | 0.214** | 0.424*** | 0.594*** | 0.326*** |
| | (0.095) | (0.138) | (0.086) | (0.068) |
| Age$^2$ | -0.003* | -0.007** | -0.010*** | -0.005*** |
| | (0.002) | (0.003) | (0.002) | (0.001) |
| Participated all games | 0.742*** | 0.745*** | 0.921*** | 0.938*** |
| | (0.187) | (0.086) | (0.124) | (0.087) |
| Market value in euro | 0.062*** | 0.106*** | 0.079*** | 0.102*** |
| | (0.008) | (0.013) | (0.011) | (0.018) |
| Zero market value | -1.238*** | -0.471** | -1.030*** | -1.173*** |
| | (0.260) | (0.225) | (0.153) | (0.106) |
| Player changed team during season | 0.147 | 0.000 | -0.124 | 0.084 |
| | (0.121) | (0.127) | (0.094) | (0.074) |
| Club position | -0.060 | -0.019 | -0.029 | 0.030 |
| | (0.027) | (0.014) | (0.022) | (0.011) |
| Player position effects | Included | Included | Included | Included |
| Club effects | Included | Included | Included | Included |
| Season effects | Included | Included | Included | Included |
| Constant | 0.458 | -4.195 | -4.786 | -2.118 |
| | (1.313) | (1.829) | (1.155) | (0.937) |
| Observations | 1,268 | 811 | 2,935 | 2,770 |
| R-squared | 0.201 | 0.198 | 0.252 | 0.268 |

Robust standard errors in parentheses.

*** p<0.01,

** p<0.05,

* p<0.1

examine the dramatic change in fans', authorities', and other players' attitudes towards Ukrainian and Russian football players after the Crimea crisis. They discuss pressure on Ukrainian soccer players in Russia [51], attitude changes of Ukrainian players towards Russia [52], or the reaction towards the transfer of Rakitsky, a Ukrainian soccer player, to a Russian top club [53].

Thus, it is reasonable to assume that the Crimean crisis resulted in a stronger effect than the effects found in previous studies.

We tested if the size of the effect is driven by outliers. To test this, we conducted an additional analysis winsorizing the continuous variables (average minutes played, player value, and age). We examined the 1st and 99th percentiles, and the 2nd and 98th percentiles. The effect is robust. For the Ukrainian soccer league, the results are robust with respect to the effects magnitude. The robustness test shows a slightly higher value and stays at the same 10% significance level. Finally, we test if the effect size varies with age or player quality. For this, we included an interaction term with a dummy for a "star" player. We defined "star" player as having a market value either higher than 1 million euros or higher than 2 million euros. The effect size is robust. The interaction, however, is not significant. Thus, there is no evidence that the Crimean crisis had a different effect on star players and non-star players in Russia. We are unable to test this for the Ukrainian league as there are no Russian star players both before and after 2014 in our sample.

Model 2 (Table 3) shows that there is also discrimination against Russian players in Ukraine after the crisis. Interestingly, Russian players are listed only in Crimean clubs after the crisis, but they're playing less.

One can assume that the negative coefficient for interaction term of Ukrainian player and crisis in Model 3 (Table 3) reflects not only the reduced time of Ukrainian players in the Russian league, but also increased Russian player playtime. Therefore, we tested an additional model with an interaction term for Russian player and crisis. The coefficient is negative but not statistically significant.

The coefficients for the control variables are as expected. Age has an inverted U-shape relationship with minutes played; the maximum playing time is for players between 30-35 years. The market value has a positive and significant effect. Players with a higher market value have a higher playing time.

### Hockey

Table 4 shows the regression estimates for the hockey leagues in Ukraine and Russia.

In Model 1 we compare the games played by Ukrainians with the games played by foreigners in Russia. In Model 2 we compare the games played by Russians to foreigners in Ukraine. In Model 3 we include both Russians and the foreign players as a control group. In Model 4 we include both Ukrainians and the foreign players as a control group.

In contrast to the results from soccer, we find that Ukrainian hockey players in Russia do not play significantly less after the Crimea crisis (Models 1 and 3), whereas Russian players play less in Ukraine (Model 2 and 4). The results are robust regardless of the control group. The control variables for the Russian league are as expected. Both age and $age^2$ of the player have the expected value. Additionally, scoring goals or assisting a team player to score goals have a positive impact. The control variables for the Ukrainian league show that several factors within the league changed, e.g., the number of teams. Both age and $age^2$ are not significant. However, the number of assists and the goals scored have the expected signs and significance value in Model 4.

## Discussion and conclusion

The Crimean crisis led to a diplomatic turning point between Russia and Ukraine culminating in a civil war in the Donbass region. The depiction of the annexation could not be more different in the two countries. In the Ukraine, the annexation is evaluated as a threat to the Ukraine as a sovereign nation. This view is supported by many western governments and international

**Table 4. Russian and Ukrainian hockey leagues.**

| | Dependent variable: Matches played | | | |
|---|---|---|---|---|
| **Model** | **(1)** | **(2)** | **(3)** | **(4)** |
| **League country** | **Russia** | **Ukraine** | **Russia** | **Ukraine** |
| **Control** | **Foreigners** | **Foreigners** | **All players** | **All players** |
| Ukrainian player | 10.632** | | 8.581* | |
| | (4.198) | | (4.373) | |
| Ukrainian player · After crisis | 5.140 | | 6.724 | |
| | (5.879) | | (6.516) | |
| Ukrainian player · After crisis · Distance to Crimea | 0.001 | | -0.222 | |
| | (0.574) | | (0.702) | |
| Russian player | | 2.348*** | | -0.866* |
| | | (0.515) | | (0.394) |
| Russian player · After crisis | | -4.217*** | | -4.837** |
| | | (1.193) | | (1.744) |
| After crisis | 3.381*** | 14.340*** | 3.544*** | 14.470*** |
| | (1.135) | (3.297) | (0.564) | (1.103) |
| Distance to Crimea | -1.024 | | 4.711** | |
| | (2.395) | | (1.745) | |
| Age | 5.234*** | 0.295 | 3.811*** | -0.436 |
| | (1.308) | (1.583) | (0.544) | (0.640) |
| Age$^2$ | -0.087*** | -0.004 | -0.065*** | 0.005 |
| | (0.023) | (0.028) | (0.010) | (0.011) |
| Plus Minus | -0.290*** | -0.504** | -0.352*** | -0.628*** |
| | (0.079) | (0.216) | (0.091) | (0.075) |
| Goals | 1.102*** | 0.135 | 1.152*** | 0.392** |
| | (0.130) | (0.273) | (0.088) | (0.132) |
| Assists | 0.951*** | 0.885*** | 1.077*** | 0.571*** |
| | (0.085) | (0.226) | (0.100) | (0.117) |
| Participated all games | 3.777** | 2.531 | 6.333*** | 5.055*** |
| | (1.466) | (4.018) | (1.142) | (1.571) |
| Player position effects | Included | Included | Included | Included |
| Club effects | Included | Included | Included | Included |
| Season effects | Included | Included | Included | Included |
| Constant | -62.190*** | -7.912 | -45.193*** | 1.666 |
| | (16.843) | (21.214) | (7.073) | (11.332) |
| Observations | 1,448 | 109 | 4,730 | 560 |
| R-squared | 0.599 | 0.612 | 0.542 | 0.596 |

Robust standard errors in parentheses.

*** $p < 0.01$,

** $p < 0.05$,

* $p < 0.1$

organizations; however, in general, in Russia the annexation is evaluated as a natural process which sooner or later was bound to happen.

We use sports data to examine how companies and employees responded to the crisis. It provides us with a quasi-experimental setting with an exogenous shock to address the issue of

political discrimination of employees. Moreover, the data is helpful as it allows us to identify whether employees are still employed and to what extent they are employed.

In Ukraine, the participation of Russian employees is severely affected by the annexation. In soccer and hockey, Russians players participate significantly less than before the crisis. However, Russian soccer players in the Ukraine after the crisis do not suffer from different treatment or behave differently themselves. [54] explanation "that political and economic forces [. . .] give(n) rise and perpetuated segmented labor markets" is a reasonable explanation for Russian employees in the Ukraine. The political and economic forces were started or exacerbated by the Crimean crisis and resulted in discrimination against general employment. The results for the Ukrainian leagues support the taste-based discrimination theory after the crisis, however there was no significant discrimination before the crisis.

The results show substantial discrimination after the annexation. When comparing this coefficient with other studies that analyze discrimination (e.g., in labor market or social integration studies) it is important to note that the occupation of the Crimean peninsula was the front runner for a civil war between Ukraine in Russia. Anecdotal evidence shows extreme change in behavior. Yevhen Seleznyov, an Ukrainian soccer player, moved to Kuban, Russia, from Dnipro, Ukraine, on February 25, 2016. He scored his first goal for his new club and participated in all games. In May 2016, however, he terminated the contract with the club by mutual agreement. His new teammate Taras Stepanenko (FC Shakhtar Donetsk) interpreted the move as follows: "Zhenya [Seleznev] realized his mistake, returned to Ukraine and is ready to work in the interests of his state." For the complete interview see https://www.sports.ru/football/1040515605.html.

In Russia, the participation of Ukrainian employees significantly decreased only in one of our target industries (soccer). In the other industry (hockey) the participation of Ukrainian employees is stable.

The results from both industries show that the employment conditions for Ukrainians in Russia and for Russians in Ukraine statistically significantly worsened since the Crimean crisis. Because the countries share a border and used to have close, although problematic, economic and political relations, our results show that currently collaboration is difficult and leads to economic loss for all involved parties.

Future research could investigate whether a similar development happened after the Crimean crisis in 1992. Another interesting research question would be to analyze whether Russian employees chose to leave the Ukraine after the crisis or whether the contracts of the Russian employees were canceled by their employers.

## Supporting information

**S1 Table. Russian and Ukrainian soccer—distance to Crimea excluded.**
(PDF)

## Author Contributions

**Conceptualization:** Iuliia Naidenova, Cornel Nesseler, Petr Parshakov, Aleksei Chusovliankin.

**Data curation:** Iuliia Naidenova, Cornel Nesseler, Petr Parshakov, Aleksei Chusovliankin.

**Formal analysis:** Iuliia Naidenova, Cornel Nesseler, Petr Parshakov.

**Methodology:** Cornel Nesseler.

**Project administration:** Cornel Nesseler.

**Supervision:** Iuliia Naidenova, Cornel Nesseler.

**Validation:** Iuliia Naidenova.

**Visualization:** Iuliia Naidenova, Petr Parshakov.

**Writing – original draft:** Iuliia Naidenova, Cornel Nesseler, Petr Parshakov.

**Writing – review & editing:** Iuliia Naidenova, Cornel Nesseler, Petr Parshakov.

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
