## [Decision Letter · Decision Letter 0]

26 May 2020

PONE-D-20-05675

After the Crimea crisis: Employee discrimination in Russia and Ukraine

PLOS ONE

Dear Mr Nesseler,

Thank you for submitting your manuscript to PLOS ONE. After careful consideration, we feel that it has merit but does not fully meet PLOS ONE’s publication criteria as it currently stands. Therefore, we invite you to submit a revised version of the manuscript that addresses the points raised during the review process.

I believe that there is some merit to this manuscript. The use of a quasi-experiment (the outbreak of a conflict) is an interesting way of estimating discrimination.

The reviewers raise a number of concerns that must be addressed.

First, the manuscript needs to better integrate the literature so as to make its point more clearly. R2 provides some useful advice on this score.

Second, both R1 and R2 raise concerns about the empirical models, what they mean and whether they are appropriate. I find this to be the most serious problem for the manuscript. Both R1 and R2 raise issues with selection effects, proper identification and basic coding issues in the data.

Third, the manuscript needs very careful editing. I will likely solicit the advice of both R1 and R2 (and perhaps another reviewer) if you decide the revise the manuscript.

We would appreciate receiving your revised manuscript by Jun 06 2020 11:59PM. To enhance the reproducibility of your results, we recommend that if applicable you deposit your laboratory protocols in protocols.io, where a protocol can be assigned its own identifier (DOI) such that it can be cited independently in the future. For instructions see: http://journals.plos.org/plosone/s/submission-guidelines#loc-laboratory-protocols

We look forward to receiving your revised manuscript.

Kind regards,

Rick K. Wilson, Ph.D.

Academic Editor

PLOS ONE

Journal Requirements:

'NO. The funders had no role in study design, data collection and analysis, decision to publish, or preparation of the manuscript.'

4. We note that Figure 2 in your submission contains map images which may be copyrighted.

We require you to either (a) present written permission from the copyright holder to publish this figure specifically under the CC BY 4.0 license, or (b) remove the figures from your submission:

b. If you are unable to obtain permission from the original copyright holder to publish this figure under the CC BY 4.0 license or if the copyright holder’s requirements are incompatible with the CC BY 4.0 license, please either i) remove the figure or ii) supply a replacement figure that complies with the CC BY 4.0 license. Please check copyright information on all replacement figures and update the figure caption with source information. If applicable, please specify in the figure caption text when a figure is similar but not identical to the original image and is therefore for illustrative purposes only.

Reviewers' comments:

Reviewer's Responses to Questions

**Comments to the Author**

1. Is the manuscript technically sound, and do the data support the conclusions?

Reviewer #1: No

Reviewer #2: Partly

Reviewer #3: Yes

2. Has the statistical analysis been performed appropriately and rigorously? 

Reviewer #1: No

Reviewer #2: No

Reviewer #3: Yes

3. Have the authors made all data underlying the findings in their manuscript fully available?

Reviewer #1: Yes

Reviewer #2: Yes

Reviewer #3: Yes

4. Is the manuscript presented in an intelligible fashion and written in standard English?

Reviewer #1: No

Reviewer #2: Yes

Reviewer #3: Yes

5. Review Comments to the Author

Reviewer #1: the topic that you address in your paper is certainly important. I like the idea to use a political shock to identify changes in employment outcomes and I particularly like the idea to use sports data. However, I have a couple of major and minor points that need to be addressed before the paper can be considered suitable for publication.

First, I have no idea how you arrive at your interpretation of the coefficients. My reading of the tables suggests that none of the relevant coefficients is statistically significant. What do the result tables contain? Is it the standard errors of the coefficients or the t-values? Did you cluster the standard errors by year/season? Since time elapsed may play a role, this should be done.

Second, yellow cards in soccer can be the result of either player or referee behavior. Since you cannot distinguish between the two sources, this test cannot produce meanungful results (are players more aggressive after the annexion or do referees punish players from the other country more harshly?)

Third, the number of observations in your descriptives tabels is far larger than the number of observations in the estimations. Why do so many player-season-observations disappear?

Fourth, you state that some Ukrainian hockey teams disappeared after the annexion. Are these teams that previously employed a large or a low number of Russian players? Is there any selection going on with the disappearing teams?

Fifth, what are the eight positions on hockey? Hockey teams typically have six players on the ice - a goalie, two/three defenders and two/three forwards. If you distinguish four positions in football, then three is certainly enough for hockey.

Sixth, I simply cannot believe that over a period of seven seasons there were 54 different clubs playing in the Russian and 53 in the Ukrainian football league (this is what the numbers in your decriptive table seem to suggest).

Seventh, to what extent does the disappearance of Russian players from the Ukranian football league affect your estimations? Have you checked the robustness of your findings?

Eight, your map seems to suggest that Crimea hosts two clubs only. Are these both football or hockey clubs? What happens to your results if you estimate your model with/without these two clubs or include a dummy to distinguish these clubs from the rest?

Finally, although not a native speaker myself, I find the English rather poor and the paper poorly edited.

Reviewer #2: Summary

The authors of this paper set up an interesting natural experiment in which they use sport as a setting to test for discrimination in the context of recent geopolitical strife in the area. Overall, I think there could be some use from this study and interesting implications. However, the paper requires considerable editing and is a bit clunky throughout. I detail my concerns with that below. There is also some question as to whether or not the effect of interest is truly being identified. Based on my readings of the models, I am not confident that the discrimination effect is the primary way one could explain the results at hand. I think some additional robustness checks are necessary, and possibly some econometric specifications changed to do so. I detail my concerns in my comments below.

Introduction and Background

-The introduction is a bit disjointed and weaves back and forth between rote literature review and specifics about the results of the current study. This could use some work such that the literature frames the problem at hand and the hypotheses of the authors. One example is on Page 4, where the results of the study are discussed, then there is just some discussion of Becker’s theory of discrimination. It continues on with some information about Levitt’s discussion of economic discrimination, but doesn’t really tie the disagreement/difficulty directly to what this paper does (is it supposed to disentangle these two theoretical approaches? If so, it doesn’t give any indication that it does). And much of this seems out of order. Further, on Page 2, last paragraph, there is a list of papers that find influence on perception of groups, but this is not well tied into the current paper or used to frame any hypotheses. Some work on integration here is needed.

Empirical Strategy and Estimations

-Substantially more needs to be provided about the DiD approach, assumptions behind this identification strategy, and so on. Most glaringly, it is unclear what the supposed control is: you are looking at the difference in the change of play time among Ukranian players in Russia (and/or Russian players in Ukraine) as compared to what, in particular? There is a pre-post period that is obvious, but you should be explicit about the comparison group (specifically: is it all players, just Russian players in Russia and Ukranian players in Ukraine, etc?)

-Are minutes played possibly impacted by changes to the makeup of worker nationalities during this time? Specifically, fewer workers in Ukraine (or vice versa) could be a result of various things related to political conflict, including wanting (needing) to be closer to home, less economic ties across borders resulting in labor movement and concentration back home, etc. If this shift away from playing/working in a certain country is inherent in the minutes played, then you may be estimating something other than discrimination. The same goes for games played. Indeed, Figure 3 seems to show this is the case: nearly no Russians remained in the Ukranian soccer league.

-If you are not including fixed effects for players, you may also be overestimating the effect by including changes to the makeup of nationalities in each country in your measurement of participation (i.e. minutes played goes to zero in Ukraine for a Russian player if that player moves to a team in Ukraine). This should at least be discussed to assuage concerns from readers about this issue. Although this could indicate hiring discrimination, there is little way to identify the causal effect here (as there are many other things that would seem likely to cause labor movement during geopolitical strife).

-You note that there is a UKR*crisis interaction, but there is no mention of a RUS*crisis interaction. This seems important, as an effect could be underestimated by only looking at, say, reduced play time for Ukranian players in a Russian league. If that play time is increasing Russian player play time (rather than just going to players of some other nationality), then the effect would be the combination of reduced Ukraine player play time AND increased Russian player play time in the Russian league, for example.

Results and Discussion

-It would be useful to flag statistical significance in Table 3, 4, 5, and 6. Also should include a note as to what is in the parentheses under each coefficient estimate (I am assuming it is the standard errors, but it is not clear).

-When you note the marginal effect is 52% (Page 14), please put this in language that relates to the units your effect is reported in (52% decline in minutes played? That seems like an enormous effect – is this believable?). I want to be sure I’m interpreting this correctly.

-It would be good to see an average discrimination effect without the inclusion of the miles away from Crimea coefficient estimate. Sometimes interpreting these with all the additional interactions can get confusing. This should be easy to do. This might be particularly important in the context of Table 4, where there are some competing effects of the UKR*crisis and the UKR*crisis*Distance interactions.

-Are penalty minutes measuring discrimination by referees or increased aggression among Ukranian players, given the geopolitical situation? This would be hard to disentangle; however, it should at least be addressed by the authors as a possible confound. The authors note “there is no evidence that Russian hockey players in Ukraine behavior more violently…” but it is not clear how the authors came to this conclusion (and it is about Russian players, rather than Ukrainian players – so I am confused here). The discussion, as with the introduction, needs some major organization fixed and some tightening up in the transitions so the reader does not get confused what model(s) you’re referring to from sentence to sentence.

Reviewer #3: This paper examines the issue of employee discrimination after a political crisis: the annexation of Crimea. The annexation, which resulted in a political crisis in the Russian-Ukrainian relations, is a setting which allows them to test if a bilateral political issue caused employee discrimination. They use a quasi-experimental approach to examine how the political crisis influenced participation and behavior in major sports leagues in Russia and Ukraine. The results show that the employment conditions significantly worsened since the Crimea crisis started.

In the introduction, I think the author does a nice job explaining the importance of the annexation of the Crimea had a significant impact on the employment situation for Ukrainian and Russian employees living in the country of the other.

This article has well structure and clearly description, and it is attractive and academically significant for researchers in the employee discrimination field.

I am very glad the authors wrote this essay.

6. PLOS authors have the option to publish the peer review history of their article (what does this mean?). If published, this will include your full peer review and any attached files.

Reviewer #1: No

Reviewer #2: No

Reviewer #3: No

---

## [Decision Letter · Decision Letter 1]

4 Sep 2020

PONE-D-20-05675R1

After the Crimea crisis: Employee discrimination in Russia and Ukraine

PLOS ONE

Dear Dr. Nesseler,

Thank you for submitting your manuscript to PLOS ONE. After careful consideration, we feel that it has merit but does not fully meet PLOS ONE’s publication criteria as it currently stands. Therefore, we invite you to submit a revised version of the manuscript that addresses the points raised during the review process.

I appreciate the revised version of this manuscript. You have been careful in addressing the concerns of the reviewers. You’ll note that both R1 and R3 are satisfied with the revision. R2, however, still has some concerns and I share them.

First, I agree with R2 (and R1’s comments on the first draft) that the yellow card analysis has too many confounds to be meaningful. Some readers might want to know whether you had these data, so you might put the analysis in the Supporting Information with a note that you explored this, but refrained from focusing on it because of unobserved confounds.

Second, I am also bothered by the effect size (R2’s point 7). Is this simply an artifact of the analysis?  That is, is there something odd going on with the econometrics, or is this a true effect?  This is critical to address.

I would like to see a revision and have you focus on convincing me that the effect size is not an artifact of the estimation strategy.

We look forward to receiving your revised manuscript.

Kind regards,

Rick K. Wilson, Ph.D.

Academic Editor

PLOS ONE

Reviewers' comments:

Reviewer's Responses to Questions

**Comments to the Author**

1. If the authors have adequately addressed your comments raised in a previous round of review and you feel that this manuscript is now acceptable for publication, you may indicate that here to bypass the “Comments to the Author” section, enter your conflict of interest statement in the “Confidential to Editor” section, and submit your "Accept" recommendation.

Reviewer #1: All comments have been addressed

Reviewer #2: (No Response)

Reviewer #3: All comments have been addressed

2. Is the manuscript technically sound, and do the data support the conclusions?

Reviewer #1: Yes

Reviewer #2: No

Reviewer #3: Yes

3. Has the statistical analysis been performed appropriately and rigorously? 

Reviewer #1: Yes

Reviewer #2: No

Reviewer #3: Yes

4. Have the authors made all data underlying the findings in their manuscript fully available?

Reviewer #1: Yes

Reviewer #2: No

Reviewer #3: Yes

5. Is the manuscript presented in an intelligible fashion and written in standard English?

Reviewer #1: Yes

Reviewer #2: Yes

Reviewer #3: Yes

6. Review Comments to the Author

Reviewer #1: The authors have careffuly addressed all the points I have raised in my report. Apart from a couple of typos I have nothing to complain about.

Reviewer #2: I appreciate the authors addressing many comments in turn in my previous review. However, I am having difficulty finding the magnitude of the effect in the paper to be believable without additional evidence and explanation. I lay out remaining concerns below and note why I think the econometric model is not properly specified:

1) Given the acknowledgement that it is not possible to distinguish between player behavior resulting in more cards vs. bias and discrimination causing more cards, the contribution of this portion of the analysis seems rather low. Given the length of the empirical analysis – and need for more clarification on the models – I would suggest focusing on the play time variables alone, and removing the yellow card analysis.

2) The taste based theory is being used to drive the question; however, the authors need to extend this into the “minutes played” portion of the analysis. As is, the way it is described refers to hiring. The authors ignore all changes to employment, and only look at minutes. This can present a disconnect for readers less familiar with this economic approach to discrimination.

3) The other issue that comes up with Becker’s theory here is that if the purported discrimination takes place across the board, then this does not put anyone at a disadvantage. The inability for markets to drive out discrimination has long been a criticism of the Beckerian approach. Are teams that play even fewer foreign players actually worse off?

4) It is not clear why Russia seeing annexation as a “natural process” is in any way counter to either discrimination or aggressive behavior. The authors need to explain their reasoning here more.

5) Make sure to be clear that you restrict sample to players appearing before and after. This is still not completely clear in the methods.

6) The choice of random effects over fixed effects for players being justified by whether the coefficients of interest are statistically significant is seriously troublesome. Did you run a Hausman test to defined your random effects choice? I have a strong feeling this is what’s driving the enormous effect size. You’re likely attributing differences in play time across players to differences within player, and claiming this is the size of the effect. If the authors think FEs for players overly control for things they seek to estimate, they need a very convincing case as to why and what the structure of that choice is. As it is, there is not a convincing case (the transfrmarkt data is useful as a control, but it’s then not clear why the effect size would change so much with FEs in lieu of this labor valuation data).

7) The effect size is just not believable and is in the range of 10 or more times the magnitude that other studies of discrimination like this have found. And given that the authors ignore dropouts (restricting to players observed only before/after), this would suggest their estimate of discrimination is biased downward. The combination of the huge effect size and the likely downward bias of the effect size tells me something is amiss in the data and analysis, there are a few overly influential players driving the effect, or the authors are incorrectly applying between player differences in play time to changes within player. If players are losing *half* their playtime, this would not go unnoticed and there would not need to be a regression to find that out. Given that, if there is a useful story or specific examples in the news that can help clearly identify this happening, then the authors should refer to this. If there is not, then there needs to be a serious explanation as to how and why we would expect to see an effect this large go completely unnoticed. A claim this extraordinary needs some supporting evidence, and an effect size this large would be easily picked up by any spectators or media covering games. It might also be useful to know if these are “stars” or if they’re largely low tier players with the purported discrimination in play time happening at the margin of bench players (where quality differences are not large). The latter would go more easily undetected. But the size of this effect really needs to be addressed and discussed, and I suspect it has more to do with the econometric specification than anything else.

8) It would be worth rescaling your distance from Crimea variable so that your coefficient is not 0.000.

9) It is not clear to me why the Distance to Crimea variable itself would have any significant coefficient. Are teams and their players (all of them) further from Crimea just playing more minutes overall? Otherwise, this seems like a strange result in your data.

10) To be able to see these possible effects, I think the authors need to cross-tabulate them in their summary tables. In other words: show minutes played by nationality (Russian or Ukranian) specifically for each league, on average. Then also include this before/after the crisis. I think looking at the transfrmarkt value crosstabulated in the same way would be enlightening to the reader to know what kinds of players are represented in your treated sample.

11) Also, in Table 1, you note that of 2,935 player-year observations, only 2% in Russia are Ukrainian and same for Russian players in Ukraine. If your n = 2,935 are correct, you are making inferences based on only 8 players in each league. This relates to my point in comment #6 above regarding an outsized effect of a very, very small number of observations that are actually treated in the model (nearly none of the observations fall into the treatment group).

Reviewer #3: If the author will continue to study this topic in the future, it is recommended to have longer-term data, so as to analyze whether the situation has changed after many years.

7. PLOS authors have the option to publish the peer review history of their article (what does this mean?). If published, this will include your full peer review and any attached files.

Reviewer #1: **Yes: **Bernd Frick

Reviewer #2: No

Reviewer #3: **Yes: **Jye-shyan Wang

---

## [Editor Report · Decision Letter 2]

5 Oct 2020

After the Crimea crisis: Employee discrimination in Russia and Ukraine

PONE-D-20-05675R2

Dear Dr. Nesseler,

We’re pleased to inform you that your manuscript has been judged scientifically suitable for publication and will be formally accepted for publication once it meets all outstanding technical requirements.

I am pleased with your careful attention to R2's concerns. I see this as a much improved draft and it makes a clear contribution using an interesting political intervention.

Kind regards,

Rick K. Wilson, Ph.D.

Academic Editor

PLOS ONE
---

## [Editor Report · Acceptance letter]

9 Oct 2020

PONE-D-20-05675R2 

After the Crimea crisis: Employee discrimination in Russia and Ukraine 

Dear Dr. Nesseler:

I'm pleased to inform you that your manuscript has been deemed suitable for publication in PLOS ONE. Congratulations! Your manuscript is now with our production department. 

Kind regards, 

on behalf of

Dr. Rick K. Wilson 

Academic Editor

PLOS ONE